# Proper Functions of Peroxisomes Are Vital for Pathogenesis of Citrus Brown Spot Disease Caused by *Alternaria alternata*

**DOI:** 10.3390/jof6040248

**Published:** 2020-10-26

**Authors:** Pei-Ching Wu, Chia-Wen Chen, Celine Yen Ling Choo, Yu-Kun Chen, Jonar I. Yago, Kuang-Ren Chung

**Affiliations:** 1Department of Plant Pathology, College of Agriculture and Natural Resources, National Chung Hsing University, Taichung 40227, Taiwan; op30132@gmail.com (C.-W.C.); ylchoo@dragon.nchu.edu.tw (C.Y.L.C.); ykchen@nchu.edu.tw (Y.-K.C.); 2Plant Science Department, College of Agriculture, Nueva Vizcaya State University, Bayombong, Nueva Vizcaya 3700, Philippines; jyago2002@yahoo.com

**Keywords:** appressorium, oleate, peroxisomes, tangerine, toxin, viability, virulence

## Abstract

In addition to the production of a host-selective toxin, the tangerine pathotype of *Alternaria alternata* must conquer toxic reactive oxygen species (ROS) in order to colonize host plants. The roles of a peroxin 6-coding gene (*pex6*) implicated in protein import into peroxisomes was functionally characterized to gain a better understanding of molecular mechanisms in ROS resistance and fungal pathogenicity. The peroxisome is a vital organelle involved in metabolisms of fatty acids and hydrogen peroxide in eukaryotes. Targeted deletion of *pex6* had no impacts on the biogenesis of peroxisomes and cellular resistance to ROS. The *pex6* deficient mutant (*Δpex6*) reduced toxin production by 40% compared to wild type and barely induce necrotic lesions on citrus leaves. Co-inoculation of purified toxin with *Δpex6* conidia on citrus leaves, however, failed to fully restore lesion formation, indicating that toxin only partially contributed to the loss of *Δpex6* pathogenicity. *Δpex6* conidia germinated poorly and formed fewer appressorium-like structures (nonmelanized enlargement of hyphal tips) than wild type. *Δpex6* hyphae grew slowly and failed to penetrate beyond the epidermal layers. Moreover, *Δpex6* had thinner cell walls and lower viability. All of these defects resulting from deletion of *pex6* could also account for the loss of *Δpex6* pathogenicity. Overall, our results have demonstrated that proper peroxisome functions are of vital importance to pathogenesis of the tangerine pathotype of *A. alternata*.

## 1. Introduction

Peroxisomes are single-membrane microbodies present in the cytosol of eukaryotic cells [1]. Peroxisomes contain enzymes involved in a broad range of metabolic processes, including degradation and synthesis of fatty acids [2,3,4] and generation and detoxification of hydrogen peroxide [5,6]. In addition to those common metabolic pathways, peroxisomes could have different functions in different kingdoms. Peroxisomes are required for the synthesis of cholesterol, bile acids, and plasmalogens in humans [7,8]. Peroxisomes are required for the gametophyte recognition during fertilization and the biosynthesis of phytohormones and biotin [9,10,11], and glyoxylate metabolic process [12,13] in plants. Cells must maintain proper functions of peroxisomes in order to grow and develop; malfunctions of peroxisomes could lead to deadly genetic disorders in humans [14]. In fungi, peroxisomes have been shown to be required for sexual and asexual reproduction [15], the biosynthesis of secondary metabolites (penicillin, cephalosporin, AK toxin, paxilline, and aflatoxins) [16,17,18,19], formation of Woronin bodies [20] and siderophores [21], glucose metabolism [22], biotin biosynthesis [23,24], and appressorium-mediated infection of host plants [25,26,27,28].

The phytopathogenic fungus *Alternaria alternata* is a necrotrophic pathogen, which kills host plant cells via the secretion of a host-selective toxin (HST) before colonization and gains nutrients exclusively from dead cells [29]. Several pathotypes of *A. alternata*: tangerine, rough lemon, Japanese pear, strawberry, apple and tomato, each producing a different HST, have been recognized. The tangerine pathotype of *A. alternata* produces a toxin called ACT, which disrupts membrane permeability, resulting in electrolyte leakage [30] and is highly toxic to tangerine, grapefruit, and their hybrids [29]. The ability to produce ACT by *A. alternata* has been demonstrated to be the primary attribute of pathogenicity to host plants [31,32]. Recent studies have demonstrated that cell wall-degrading enzymes (CWDEs) are also critical during *A. alternata* pathogenesis to citrus [33,34].

In addition to ACT and CWDEs, *A. alternata* must be able to detoxify reactive oxygen species (ROS) in order to achieve successful colonization within citrus tissues [35,36]. The citrus pathotype of *A. alternata* has evolved effective defense systems to thrive within the oxidative environment of necrotic tissues. ROS detoxification is a complex process that involves multiple transcription regulators and signaling pathways. Previous studies have identified several proteins and pathways that are required for ROS resistance in the pathotype of *A. alternata*. Those include the Yap1 bzip transcription factor [37,38,39], the Skn7 response regulator [40], the Tfb5 basal transcription factor II [34], the nicotinamide adenine dinucleotide phosphate oxidase (NADPH oxidase, Nox) [41,42], the nascent polypeptide-associated complex [43], the siderophore-mediated iron uptake system [44,45,46], the Hog1 mitogen-activated protein kinase (MAPK)-mediated pathway [47,48,49], the glutaredoxin and thioredoxin systems [50,51], and the major facilitator superfamily (MFS) membrane transporters [52,53]. Defects in any of the aforementioned proteins or pathways in *A. alternata* reduce cellular sensitivity to ROS and infectivity on citrus leaves. 

Comparative transcriptome analysis between wild type and an *A. alternata* mutant impaired for NADPH oxidase identified many *pex* genes, whose products are required for peroxisomal biogenesis and function. Because peroxisomes have been implicated in cellular detoxification of hydrogen peroxide [1,5,6], we hypothesized that peroxisomes may also play a role in ROS resistance in the tangerine pathotype of *A. alternata*. More than 32 peroxin (Pex) proteins have been identified to be required for peroxisome biogenesis and protein translocation [54,55]. In the present study, a *pex6* gene, which encodes a protein required for matrix protein import into peroxisomes [56,57], was characterized to play important biological and pathological roles in the tangerine pathotype of *A. alternata*, even though *pex6* plays no role in resistance to ROS.

## 2. Materials and Methods

### 2.1. Fungal Strains and Culture Conditions

The wild-type strain of *Alternaria alternata* (Fr.) Keissler used in this study as a recipient host for transformation and mutagenesis experiments was cultured from a diseased citrus leaf and has been previously characterized [37]. Fungal strains were grown on potato dextrose agar (PDA, Difco, Sparks, MD, USA) or minimal medium (MM) [58] at 28 °C with constant fluorescent light for 3–5 days for the formation of conidia. Conidia were harvested from the surface of fungal colony by flooding with sterile water and scrapping with a spatula. For toxin production, fungal strains were grown in potato dextrose broth (PDB) for 7 days. For DNA isolation, fungal strains were grown on PDA or PDB for 2–3 days. Fungal transformants were recovered from a regeneration medium (RMM) as described [58].

### 2.2. Assays for Germination of Spores, Formation of Appressorium-Like Structures and Cell Viability

Assays for conidia germination were performed by placing conidial suspensions on glass slides or citrus leaves, or in PDB in 96-well microtiter plates, and incubating in a plastic box for 6 h. The formation of appressorium-like structures was evaluated by culturing conidia in PDB or liquid MM placed in 96-well microtiter plates. The germination of spores was examined by light microscopy. Cell viability was assessed by 0.05% Evan’s blue staining in which dead cells incapable of expelling dye were stained blue and live cells remained light brown. Lipid bodies within fungal cells were stained with Nile red and examined by fluorescence microscopy. 

### 2.3. Sensitivity Tests

Sensitivity tests were evaluated by transferring fungal mycelia as a toothpick point inoculation to PDA containing a test compound and incubated under constant fluorescent light. Tested compounds included oleic acid (0.5% dissolved in ethanol), Tween 20 (a polysorbitan containing lauric acid, 0.5%), calcofluor white (CFW, 200 µM) and Congo red (CR, 50 µM). Fungal radial growth was measured at 3–4 days. Each treatment contained three replicates and experiments were repeated at least three times. Percentage of growth inhibition was calculated by dividing the comparative difference of the growth by the wild-type growth and multiplying by 100.

### 2.4. Molecular and Genetic Procedures

The *A. alternata pex6* gene (accession # OWY50363) fragment was amplified by PCR with the primers from genomic DNA prepared from wild type. Fungal DNA was isolated using a Genomic DNA Mini Purification kit (BioKit, Taipei, Taiwan). A 1.2-kb *pex6* region within open reading frame was replaced by a bacterial phosphotransferase B gene (*hyg*) cassette under control of the *Aspergillus nidulans trpC* gene promoter and terminator, which conferred resistance to hygromycin, in the genome of *A. alternata* with a split marker approach as described [58]. A 5’ or 3’ *pex6* fragment was fused with a truncated *hyg* fragment by two-round PCR with gene-specific primers (Appendix A) and transformed into protoplasts prepared from wild type using the CaCl_2_ and polyethylene glycol (PEG)-mediated method as described previously [58]. Transformants were recovered from RMM amended with 200 µg/mL hygromycin (Roche Applied Science, Indianapolis, IN, U.S.A.) and examined by PCR with a distal *pex6* primer pairing with a *hyg* primer and Southern blotting using a *pex6* or a *hyg*-specific gene probe. DNA probes used for Southern blot analysis were prepared by PCR to incorporate digoxigenin (DIG)-11-dUTP into the probe with two *pex6*-specific primers. The probe was detected by immunological assays according to the manufacturer’s recommendation (Roche Applied Science). For *pex6* complementation, a *pex6* fragment containing its endogenous promoter was amplified by primers *Pex6*_cp1 3.0 (5’-aatgcggccgcccatccatctctgcttctttcgga-3’), *Pex6*_cp2 pst1 (5’-atctgcagttaagagtacaaatcgtcatcctg-3’) and cloned into pCB1532 after endonuclease digestion (*Not*I and *Pst*I). The resultant plasmid was transformed into protoplasts prepared from a *pex6*-deficient mutant (M4) and transformants recovered on regeneration medium (RMM) containing 5 µg/mL of sulfonylurea (Chem Service, West Chester, PA, USA).

### 2.5. Virulence Assays

Fungal virulence and bioassays for toxin were conducted on detached calamondin (*Citrus mitis* Blanco) leaves. Conidia suspensions (10^5^ or 10^8^ conidia/mL) were prepared in sterile water and 10 μL applied to a spot on calamondin leaves with or without wounding. Leaf spots were wounded before inoculation with a fine needle to facilitate fungal penetration. Leaf spots treated with sterile water were used as mock controls. All treated leaves were maintained in a plastic box (12 h daily illumination, 98% relative humidity, and 26 °C constant temperature) for 3 days for lesion development. Each strain was tested on at least five different leaves and experiments were repeated at least three times.

### 2.6. Purification and Analysis of Host-Selective Toxin

The ability to produce ACT toxin by *A. alternata* strains was assessed by culturing fungi in PDB for 7 days in the dark. Cell-free culture filtrates were prepared by consecutively passing through a filter paper and a 0.45 µm polyethersulfone (PES) membrane (Thermo Fisher Scientific, Hampton, NH, USA). Bioassays for the toxicity of culture filtrates were performed by immersing the petiole of the leaf in cell-free culture filtrates in a 1.5-mL microcentrifuge tube placed in a plastic rack and incubated in a plastic box for 2 to 3 days for lesion development. The presence of ACT toxin was evidenced by the appearance of necrotic lesions along the midrib and veins. Necrotic areas were measured using Image J software available at https://imagej.nih.gov/ij/. Each treatment contained at least five leaves. Toxin bioassays were also carried out by placing 10 µL cell-free culture filtrates at the base of the petiole or directly on the surface of the leaf that was wounded by a fine needle. For chromatography analyses, culture filtrates were mixed with Amberlite XAD-2 resin (Sigma-Aldrich, St. Louis, MO, USA), eluted with methanol, and extracted with ethyl acetate as described [30]. After air-drying in a pre-weighted centrifuge tube, the amount of ACT was calculated and normalized to the dry weight of mycelium. ACT was dissolved in methanol and separated in a 60 F254 silica gel plate using benzene/ethyl acetate/acetic acid (50:50:1, *v*/*v*), visualized under a hand-held UV illuminator, and photographed. For high performance liquid chromatography (HPLC), ACT toxin was dissolved in methanol: Milli-Q (MQ) water: acetic acid (60:40:1, *v*/*v*) solution and separated by an Ascentis C18 column (5 µm particle size silica, 4.6 i.d. × 250 mm) attached to a Agilent 1100 series HPLC (Santa Clara, CA, USA). The mobile phase consisted of methanol: MQ water: acetic acid (60:40:1, *v*/*v*) at a flow rate set at 1.0 mL/min and recorded the absorbance at 290 nm was recorded. The solution eluted at 4.1–4.4 min and 7.4–7.9 min was collected and used for leaf necrotic assays by placing the elutes on the calamondin (*Citrus mitis* Blanco) leaves as described above. The identity of ACT was also confirmed by liquid chromatography/tandem mass spectrometry (LC-MS/MS).

### 2.7. Microscopy 

Germination of conidia was examined with a Nikon Optiphot 2 Microscope (Tokyo, Japan). Lipid bodies within fungal cells were stained with 2.5 μg/mL Nile red (Sigma-Aldrich) solution for 20 min at room temperature and examined by a ZOE Fluorescent Cell Imager (Bio-Rad, Hercules, CA, USA) microscope equipped with the 530-nm excitation and 635-nm emission wavelength filters. Transmission electron microscopy (TEM) was performed using a JEOL JEM-1400 series 120 kV Transmission Electron Microscope (Jeol, Tokyo, Japan). Samples were treated with 2.5% glutaraldehyde in 0.1 M phosphate buffer (pH 7.2) and 1% osmium tetraoxide, and immersed in LR White Resin after being dehydrated with an ethanol series. Samples were cut to thin slices, stained with uranyl acetate and lead citrate, and examined. Photographs were taken with an Gatan Orius SC 1000B bottom mounted CCD-camera (Gatan Inc., Pleasanton, CA, USA). For penetration studies, leaf samples inoculated with fungal strains were sectioned to thin slices by hand using a sharp razor blade, stained with lactophenol cotton blue (Sigma-Aldrich), and examined by light microscopy.

### 2.8. Statistical Analysis

Unless otherwise indicated, all experiments were conducted three times with at least three replicates. The significance of treatments was analyzed by one-way ANOVA (analysis of variance) and treatment separated by post-hoc Tukey HSD (honestly significant difference) test (*p* < 0.05).

## 3. Results

### 3.1. Identification and Mutation of Pex6 in the Tangerine Pathotype of A. alternata

The *A. alternata pex6* gene (accession # OWY50363) encoding a peroxisomal protein was originally identified from comparative transcriptome analysis between wild type and a fungal strain carrying both *noxA* and *noxB* mutations. The *pex6* gene was found to have a 4481-bp open reading frame (ORF), which encodes a polypeptide of 1441 amino acids after spicing out two small introns of 57 and 89 bp. *Pex6* was found to contain two AAA+ (ATPase associated with various cellular activities) domains. To investigate the biological functions of *Pex6*, the coding gene was deleted by replacing with a hygromycin resistance gene (*hyg*) cassette under control of the promoter and the terminator of *trpC* (Figure 1A). Two overlapping but truncated *hyg* fragments fused with 5’ or 3’ end of *pex6* were transformed into protoplasts prepared from the wild-type strain of *A. alternata*, leading to the identification of two independent transformants (designated *Δpex6*-M4 and M9). Both M4 and M9 strains had a single *hyg* insertion at the *pex6* locus based on Southern blotting analyses using a *pex6* (Figure 1B) or a *hyg* (Figure 1C) gene-specific probe. A 1.2-kb region of *pex6* was replaced with a *hyg* cassette in the genome of both M4 and M9 mutants.

### 3.2. Pex6 is Required for Hyphal Growth and Conidia Germination

*Δpex6* strains cultured on potato dextrose agar (PDA) for 7 days reduced growth by 40% compared to wild type (Figure 2A). On minimal medium (MM), growth reduction in *Δpex6* was more drastic, reducing by as much as 60 % compared to wild type. The CP4 strain, which was identified from the M4 strain after being transformed with a functional copy of *pex6* with its endogenous promoter, displayed wild-type growth on PDA and MM. Both wild type and CP4 generated obclavate, melanized, muriform conidia divided by horizontal and vertical septae (Figure 2B). *Δpex6* strains produced ample conidia at levels comparable to those of wild type and CP4. Many of conidia produced by *Δpex6* failed to germinate, reducing by 50% compared to those produced by wild type and CP4 (Figure 2C).

### 3.3. Pex6 Impacts the Formation of Appressorium-Like Structures

Conidia of *A. alternata* germinated and produced nonmelanized enlargement at the end of germ tubes (resembling an appressorium) when culturing on glass slides or 96-well microtiter plates (Figure 2D) or after being inoculated on detached calamondin leaves. Quantitative analysis revealed that both wild type and CP4 conidia germinated efficiently and more than 55% of germ tubes produced appressorium-like structures when culturing in MM on microtiter plates (Figure 2E). *Δpex6* conidia germinated and less than 30% of germ tubes developed to form appressorium-like structures. When culturing in PDB, *Δpex6* formed appressorium-like structures at rates which were not significantly different from wild type (Figure 2E). 

### 3.4. Deletion of Pex6 Leads to Low Conidia Viability

The viability of fungal cells was evaluated after staining with 0.05% Evan’s blue (Figure 3A). More than 50% of conidia produced by *Δpex6* grown on MM were stained blue, indicative of cell death (Figure 3B). Conidia collected from *Δpex6* grown on PDB or placed in distilled water for 2 h (data not shown) also reduced viability by ~15% compared to those of wild type and CP4. There was no significant difference in melanin purified from cell walls of wild-type and *Δpex6*.

### 3.5. Pex6 is Involved in Lipid Metabolism and Maintenance of Cell-Wall Integrity

*Δpex6* reduced radial growth on PDA by ~40% compared with wild type. Compared to growing on PDA alone, *Δpex6* displayed less than 15% growth reduction on PDA amended with oleic acid or Tween 20 (a polysorbitan containing lauric acid) (Figure 4A), indicating adding oleic acid or Tween 20 could partially but significantly restore *Δpex6* growth deficiency (Figure 4B). Percentage of *Δpex6* growth reduction in relation to wild type decreased on PDA amended with increasing the amount of oleic acid (Figure 4C). *Δpex6* reduced radial growth by nearly 50% and 80% on PAD amended with calcofluor white (CFW) and Congo red (CR), respectively. The addition of methanol (2.5%), FeCl_3_ (0.2 mM), EDTA (0.05 mM) or SDS (0.02%) into PDA had no effects on the growth reduction of *Δpex6*. Sensitivity tests conducted on MM revealed that *Δpex6* failed to efficiently utilize glucose, oleic acid, or Tween 20 as the sole carbon source (Figure 4A and data not shown). However, the addition of both glucose and oleic acid together could partially restore the growth of *Δpex6* (Figure 4A).

Staining fungi with Nile red, which selectively stains lipid droplets in cells, revealed that hyphae and conidia of both wild type and CP4 were uniformly stained red (Figure 4D). Conidia of *Δpex6* were also stained red, while most of the *Δpex6* hyphae were barely stained by Nile red. Transmission electron microscopy (TEM) revealed that the number of peroxisomes (appearing as dark spherosomes) within the wild-type and the *Δpex6* hyphae was similar (average 2 peroxisomes per cell, *n* ≥ 40) (Figure 5A,B). TEM analyses revealed that wild-type hyphae had more lipid bodies (appearing as blight spherosomes) than *Δpex6* (Figure 5C,D). Wild-type hyphae contained an average of 4 lipid bodies per cell (*n* = 40), while *Δpex6* hyphae had an average of 2 lipid bodies per cell (*n* = 46). TEM analyses also revealed that wild type had a thicker cell wall than *Δpex6* (Figure 5E). *Δpex6* hyphae had the cell-wall width around ~0.3 µm (*n* = 30), which was significantly thinner than those of wild type (Figure 5F).

### 3.6. Pex6 is Required for Lesion Formation

Fungal virulence was assessed on detached calamondin leaves inoculated with conidial suspensions (10^5^ conidia/mL), revealing that *Δpex6* failed to induce necrotic lesion, while wild type and CP4 induced necrotic lesions three days post inoculation (dpi) (Figure 6A). Inoculation of the calamondin leaves that were wounded before inoculation with conidia (10^5^ conidia/mL) of *Δpex6* produced much smaller or completely failed to yield visible lesions 3 dpi (Figure 6B). Some of the spots inoculated with the mutants eventually developed small necrotic lesions 7 dpi (data not shown). Inoculation of citrus leaves with higher concentrations (>10^8^ conidia /mL) of *Δpex6* conidia occasionally resulted in large necrotic lesions 3–5 dpi.

### 3.7. Pex6 Deficiency Leads to Penetration Impairment

Light microscopy observation of conidia germination on the surface of citrus leaves revealed that wild-type conidia germinated, formed appressorium-like structures, and penetrated through stomata 1 dpi (Figure 7A). Microscopic lesions were observed around the penetration site 1 dpi, which continuously developed to become large necrotic lesions 3 dpi (Figure 7B). Wild type proliferated and formed interlaced hyphae on the surface of the leaves. Conidia of *Δpex6* after being inoculated onto calamondin leaves also germinated 1 dpi, despite at a lower rate compared with that of wild type. Similar to *in-vitro* assays, *Δpex6* germ tubes produced less appressorium-like structures than wild type on the surface of the leaves. *Δpex6* hyphae often bypassed stomata but some could initiate penetration. Microscopic lesions induced by *Δpex6* were sporadically observed 3 dpi. Light microscopy examination of cross section of a leaf after being inoculated with wild type or *Δpex6* revealed that wild-type penetrated efficiently, resulting in collapse of leaf tissues, in which epidermal layers were deranged and mesophyll cells disfigured (Figure 8). Wild-type hyphae were observed abundantly within mesophyll cells 3 dpi. In contrast, conidia of *Δpex6* germinated and penetrated through a stoma or wound into the epidermal layers. *Δpex6* hyphae failed to invade further into mesophyll cells. No aggregation of *Δpex6* hyphae was observed within mesophyll cells 3 dpi. Citrus leaves inoculated with 10^5^ conidia/mL of *Δpex6* rarely developed large necrotic lesions, unless higher concentrations (>10^8^ conidia/mL) of conidia were used for inoculation.

### 3.8. Pex6 Deficiency Impacts Toxin Production

Wild-type cell-free culture filtrates after being applied to calamondin leaves resulted in necrotic lesions 1 dpi (Figure 9A). Quantitative analysis of the necrotic lesions revealed that cell-free culture filtrates prepared from *Δpex6* resulted in less severe necrosis on the leaves than those prepared from wild type and CP4 (Figure 9B). Host-selective toxin (ACT) was purified and analyzed by thin-layer chromatography (TLC) also revealed that *Δpex6* produced less ACT than the wild-type and the CP4 strains at similar fungal mass (Figure 9C). HPLC analysis of purified ACT resulted in three major peaks with retention times of 4.1, 7.4, and 7.9 min (Figure 9D). Samples prepared from wild type and CP4 had greater peak areas than those from the *Δpex6* strains. Leaf necrosis assays were performed with the samples eluted from 4.1 min and 7.4/7.9 min peaks. No lesion was detected in samples purified from the 4.1-min peak (data not shown). Samples eluted from the 7.4/7.9-min peaks of *Δpex6* resulted in less severe necrosis on detached calamondin leaves compared to those collected from wild type or CP4 (Figure 9E). Based on peak areas, *Δpex6* reduced ACT production by ~40% compared to wild type. The involvement of peroxisomes in the biosynthesis of ACT was confirmed further by identifying a Peroxisomal Targeting Sequence 1 (PTS1) with conserved tripeptide sequence S/C/A-K/R/H-L in the C terminus of three enzymes: putative hydrolase (ACTT2, accession no. OWY54915.1), HMG-CoA hydrolase (ACTT3, accession no. OWY54927.1), and AKT1 (accession no. OWY56622.1) implicated in the biosynthesis of ACT.

### 3.9. Co-Inoculation of Δpex6 with Toxin Partially Restores Lesion Formation 

Pathogenicity assays revealed that calamondin leaves, after being treated with purified ACT (~12–15 µg toxin/g mycelium) at a 10-fold dilution and, 30 min later, inoculated with conidial suspensions of *Δpex6,* developed necrotic lesions that were slightly larger than those treated with toxin alone 3 dpi (Figure 10). ACT at 100- or 1000-dilution failed to result in necrotic lesions on the leaves after being applied alone or co-inoculated with conidial suspensions of *Δpex6*. Calamondin leaves inoculated with conidial suspensions prepared from wild type with or without ACT resulted in necrotic lesions 3 dpi.

## 4. Discussion

Peroxisomes (originally described as microbodies) are dynamic organelles found in all eukaryotes [1,54,55]. Peroxisomes play a wide range of metabolic functions ranging from the oxidation of fatty acids, generation and detoxification of hydrogen peroxide to the biosynthesis of secondary metabolites depending on the environment, the organism, and developmental stage of the cell [4,6,20]. In the present study, we have characterized a peroxin 6-coding gene (*pex6*) to play important biological and pathological roles in the tangerine pathotype of *A. alternata*. TEM observations reveal that mutation of this gene in *A. alternata* does not impact peroxisome biogenesis, which is consistent with the notion that *Pex6* is required for matrix protein import into peroxisomes [56,57]. Collectively, we have found that *pex6* is required for the formation of appressorium-like structures and lipid bodies, cell viability, toxin production, and pathogenicity in *A. alternata*. *Pex6* is dispensable for the formation of peroxisomes and conidial production, but is critical for germination and viability of conidia. Microscopy examination reveals that *Δpex6* fails to penetrate through the leaf mesophyll. The results indicate that *pex6* has a profound impact on penetration through stomata and hyphal elongation within citrus leaf tissue, which could result from a combination of multiple impairments. All defects seen in *Δpex6* could be completely reversed to the wild-type levels by expressing a functional copy of *pex6*, confirming that all observed phenotypes are attributed to the deletion of *pex6*. Overall, our results have demonstrated that peroxisomes are critical for developmental and physiological functions and *A. alternata* pathogenesis to citrus. 

Although *pex6* is conserved in fungi, its functions in the formation and germination of conidia and hyphal elongation vary considerably among different species. The *Δpex6* mutant displays 40 and 60% growth reduction compared to wild type grown on PDA and MM (containing glucose as the sole carbon source), respectively. It seems that loss of *pex6* has more drastic effects on the growth of the tangerine pathotype of *A. alternata* than other fungi. Mutation of the *pex6* homolog in *Colletotrichum lagenarium* [25] has little or no effect on radial growth. However, mutation of *pex6* or other peroxin-coding genes in the rice blast fungus *Magnaporthe oryzae* often results in growth retardation [26,59,60]. Deletion of the *pex6* homolog in the Japanese pear pathotype of *A. alternata*, which produces the host-selective AK toxin, reduces radial growth by ~20% on MM containing glucose as the sole carbon source [16]. *Pex6* is required for conidiation in *M. oryzae* and the Japanese pear pathotype of *A. alternata* and dispensable in the tangerine pathotype of *A. alternata* and *C. lagenarium*. Moreover, *pex6* plays a role in conidia germination in the tangerine pathotype, but not in the Japanese pear pathotype of *A. alternata*. Those results indicate that *pex6* has very different physiological and developmental functions in fungi, even between two very closely related pathotypes.

*Pex6* may play a role in maintaining the chemical composition of fungal cell wall in the tangerine pathotype of *A. alternata*. *Δpex6* increases sensitivity to cell-wall-damaging agents (Congo red and calcofluor white), suggesting the involvement of *pex6* in the maintenance of cell wall integrity. This is confirmed further by electron microscopy, revealing that the *Δpex6* has much thinner cell wall than wild type. The *pex6*-deficiency mutant of *M. oryzae* also increases sensitivity to calcofluor white [61]. *Pex6* also plays a vital role on cell viability in the tangerine pathotype; many of conidia produced by *Δpex6* fail to germinate. Evan blue’s staining reveals that, in relation to conidia produced by wild type, a higher proportion of conidia produced by *Δpex6* are stained blue, indicative of dead cells. Conidia produced by *Δpex6* grown on MM have lower viability compared to those collected from PDA. Mutation of the *pex6* homolog in *F. graminearum* also leads to low long-term cell viability [28]. Thus, proper functions of peroxisomes are required for cell survival in fungi. 

Although peroxisomes have been implied to be required for detoxification of hydrogen peroxide [1,5,6], *pex6* apparently plays no roles in ROS resistance in the tangerine pathotype of *A. alternata*. This is likely due to the notion that *pex6* encodes a peroxin required for matrix protein import into peroxisomes [56,57] rather than peroxisomal biogenesis. TEM analysis also reveals no significant differences in the sizes and numbers of peroxisomes between wild type and *Δpex6*. Sensitivity assays reveal that *Δpex6* displays wild-type sensitivity to H_2_O_2_ (10 mM), *tert*-butyl hydroperoxide (1.5 mM), KO_2_ (1 mg/mL) and diethyl malonate (2.5 mM) (data not shown). Although *pex6* is not involved in resistance to oxidative stress, the roles of peroxisomes in ROS resistance warrant further investigation in the tangerine pathotype. 

Nile red is commonly used to stain lipid droplets in cells. Conidia produced by *Δpex6* show strong red fluorescence after staining with Nile red. In contrast, most of the *Δpex6* hyphae do not show red fluorescence, indicating the presence of fewer lipids in the hyphae of *Δpex6*. This is confirmed further by TEM observation. Fewer lipid bodies were observed in the hyphae than conidia of *Δpex6*, suggesting that *Pex6* is likely required for lipid mobilization from conidia to hyphae. Reduced lipid bodies in hyphae could result in slow growth, at least in part, due to the lack of sufficient energy supply in *Δpex6*. Indeed, adding oleic acid or Tween 20 promotes the growth of wild type and restores the growth of *Δpex6* on PDA. On MM containing glucose as the sole carbon source, the addition of oleic acid can promote wild-type growth, but only partially restores *Δpex6* growth. Although *Δpex6* is unable to use glucose or oleic acid as the sole carbon source effectively, *Δpex6* grow better in the presence of both glucose and oleic acid. This could be simply due to the presence of two carbon sources. The *pex6* mutants of *F. graminearum* and the Japanese pear pathotypes of *A. alternata* also are unable to effectively utilize oleic acid or Tween 80 (polyoxyethylene sorbitan monooleate) as the sole carbon source, resulting in a severe growth reduction [16,28].

The tangerine pathotype of *A. alternata* produces appressorium-like structures on glass slides and 96-well microtiter plates in water or low-nutrient medium (e.g., MM), and on the surface of citrus leaves. The role of the appressorium-like structures in relation to fungal penetration to citrus leaves remain largely unknown. On the surface of citrus leaves, wild-type appressorium-like structures are often formed near the stomata, implicating an important role in sensing stomata and penetration. Compared with wild type, *Δpex6* produces much fewer appressorium-like structures, which are often formed away from stomata. Because *Δpex6* hyphae fail to penetrate into citrus mesophyll cells and to induce large necrotic lesions, it is tempting to speculate that appressorium-like structures produced by *Δpex6* are somewhat defective and that appressorium-like structures may be required for *A. alternata* pathogenesis to citrus. The *pex6* mutants of *M. oryzae*, *C. lagenarium* and *C. orbiculare* produce deformed appressoria, which fail to infect their respective host plants [25,26,61]. The *pex6* mutant of the Japanese pear pathotype of *A. alternata* can form appressorium-like structures, but their penetration functions are compromised [16]. The co-application of AK toxin with the *pex6* mutant of the Japanese pear pathotype slightly restores the formation of penetration hyphae from appressoria but completely restores lesion formation, indicating that AK toxin is the key factor contributing to nonpathogenic deficiency of the *pex6* mutant on pear leaves. 

The roles of *pex6* in the toxin production and fungal pathogenicity are different between the tangerine and the Japanese pear pathotypes. The *pex6* mutant of the Japanese pear pathotype completely loses its ability to produce AK toxin, and its pathogenicity can be fully restored by co-applying AK toxin with conidia on pear leaves [16]. Purification and assays of ACT reveal that *Δpex6* reduces the production by ~40% compared with wild type. This could, at least in part, account for the reduction of *Δpex6* virulence on citrus leaves. However, unlike the Japanese pear pathotype, co-applying ACT toxin with conidial suspensions prepared from wild type fail to fully restore *Δpex6* virulence on citrus leaves, indicating that a reduced production of ACT by *Δpex6* is not the only factor contributing to the reduction of virulence. Because toxins can only partially restore the virulence of *Δpex6*, it appears that growth retardation, cell viability, cell wall deformation, reduced formation of appressorium-like structures, and penetration deficiency may also account for the reduction of fungal virulence.

In conclusion, peroxisomes are important organelles in all eukaryotes. Our studies provide new insights into the role of appressorium in the pathogenesis of *A. alternata* to citrus leaves and underlie the importance of peroxisome functions in fungal development, physiology, and pathogenicity. Further investigation of the roles of peroxisomes in ROS resistance and cell death will shed some light on how the tangerine pathotype of *A. alternata* responds to stress and nutrient availability during penetration and colonization on citrus leaves.

## Figures and Tables

**Figure 1 jof-06-00248-f001:**
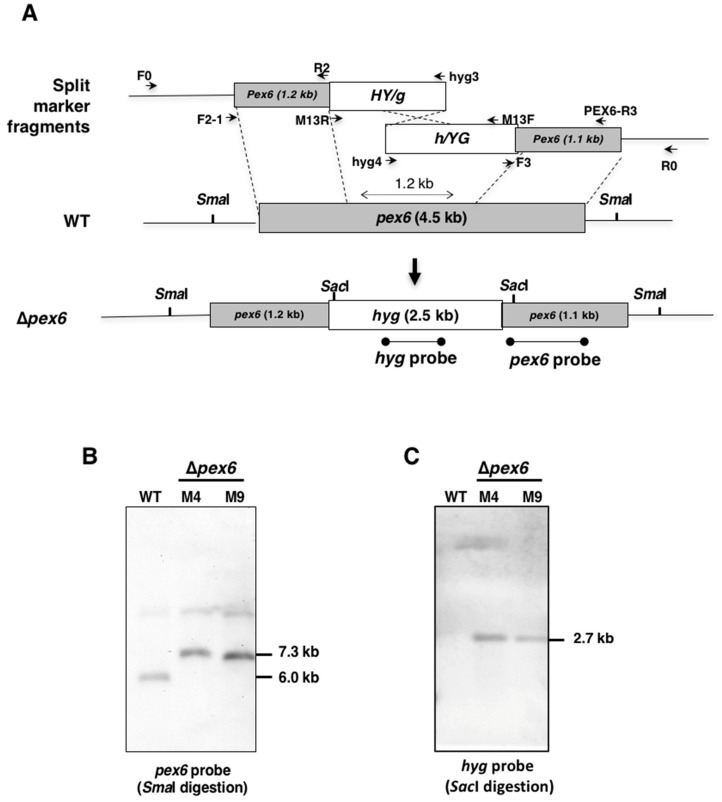
Deletion of a peroxin 6-coding gene (*pex6*) using a split marker approach in the tangerine pathotype of *A. alternata*. (**A**) Schematic depiction of gene disruption of *pex6* via a homologous recombination using truncated but overlapping DNA fragments of a hygromycin phosphotransferase-coding gene (*hyg*) under control by the *Aspergillus nidulans trpC* promoter and terminator. Split marker fragments, 5’*pex6*::5’HY/g fragment and 3’*pex6*::3’h/YG fusion fragment, were amplified with primers F2-1 pairing with hyg3 and PEX-R3 pairing with hyg4, respectively. Oligonucleotide primers used to amplify each fragment are indicated; (**B**) Southern-blot hybridization of genomic DNA prepared from wild-type (WT) and two putative *pex6* mutants (*Δpex6*-M4 and M9), in which a 1.2-kb region of *pex6* ORF was replaced with a *hyg* cassette. Fungal DNA was cleaved with *Sma*I endonuclease, electrophoresed, blotted to a nylon membrane, and hybridized with a *pex6* probe. A faint band (estimated > 10 kb), likely due to incomplete digestion, was detected in all three samples; (**C**) Southern-blot hybridization of genomic DNA digested with *Sac*I and hybridized with a *hyg* probe.

**Figure 2 jof-06-00248-f002:**
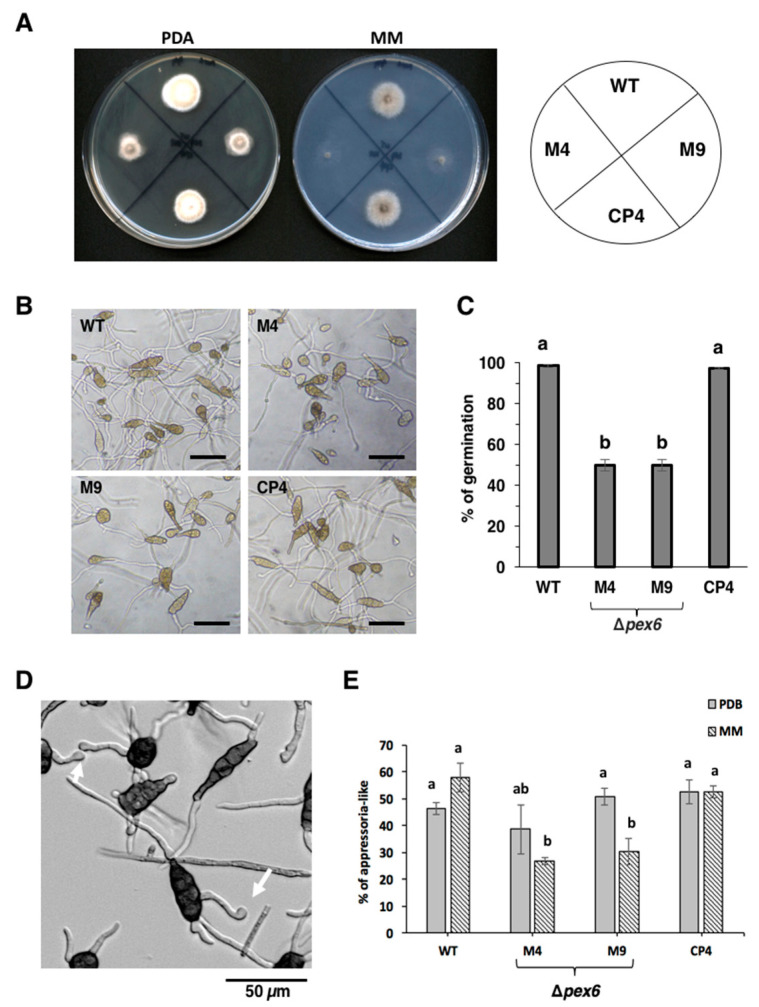
*Pex6* is required for hyphal growth, conidia germination and the formation of appressorium -like structures in *A. alternata*. (**A**) Deletion of *pex6* results in growth retardation. Fungal strains: wild type (WT), two *pex6* mutants (*Δpex6*-M4 and M9) and the complementation strain (CP4) were grown on potato dextrose agar (PDA) or minimal medium (MM) for 3 days; (**B**) Deletion of *pex6* plays no roles in conidia formation. Bar = 50 µm; (**C**) Conidia of both M4 and M9 strains germinate at rates much slower than those of wild type and CP4; (**D**) The formation of appressorium-like structures (nonmelanized enlargement of hyphal tips, indicated by arrows) by wild type on microtiter plates; (**E**) When culturing in liquid MM, *Δpex6* produces fewer appressorium-like structures on microtiter plates. Significance of difference between means was analyzed using Tukey’s HSD post-hoc test (*p* ≤ 0.05). Means (*n* = 100) followed by the same letters are not significantly different.

**Figure 3 jof-06-00248-f003:**
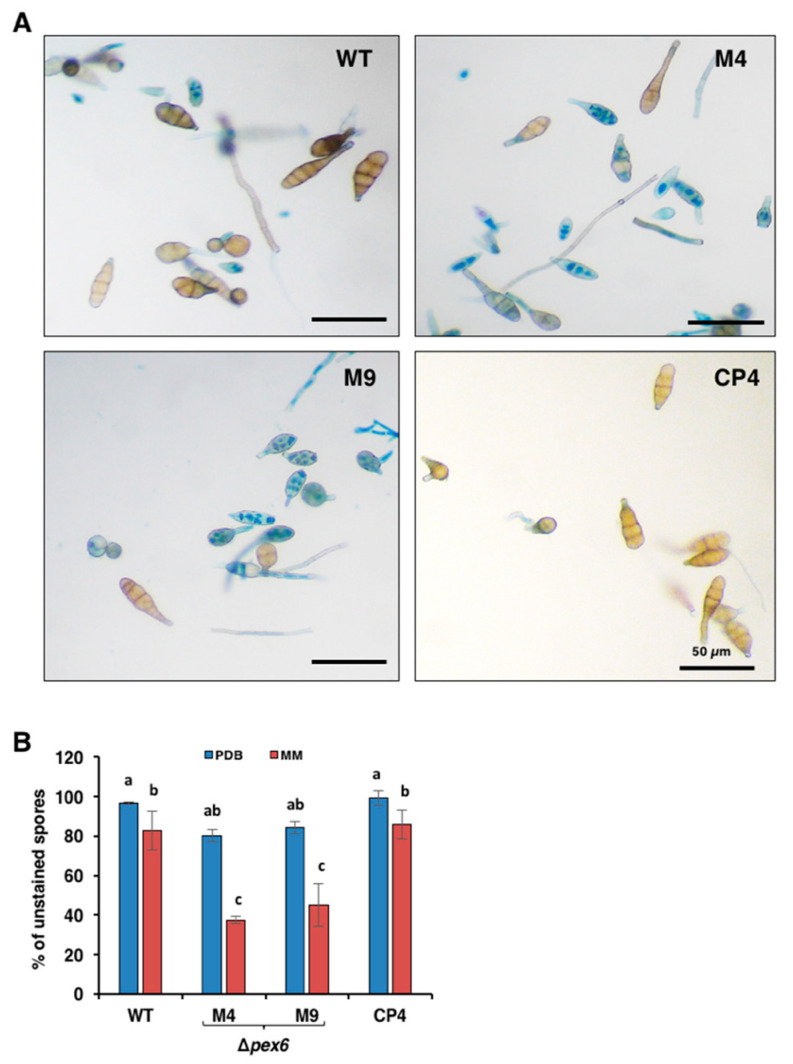
*Pex6* is required for cell viability in *A. alternata*. (**A**) Assays for the viability of conidia prepared from wild type (WT), two *pex6* mutants (*Δpex6*-M4 and M9) and the complementation strain (CP4) by Evan’s blue staining. Live cells are capable of expelling dye and remain clear, whereas dead cells fail to expel dye and are stained blue; (**B**) Percentage of unstained conidia collected from fungal strains were incubated in potato dextrose broth (PDB) or minimal medium (MM) for 7 h. Significance of difference between means was analyzed using Tukey’s HSD post-hoc test (*p* ≤ 0.05). Means (*n* = 100) followed by the same letters are not significantly different.

**Figure 4 jof-06-00248-f004:**
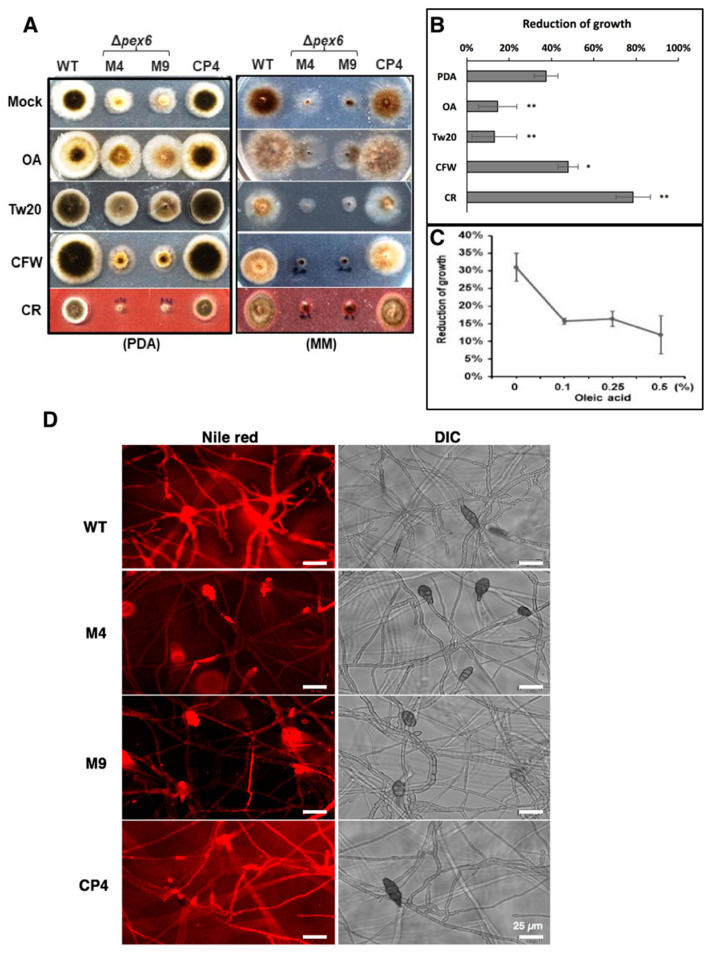
*Pex6* is involved in lipid metabolism and maintenance of cell-wall integrity in *A. alternata*. (**A**) Assays for sensitivity of wild type (WT), two *pex6* mutants (*Δpex6*-M4 and M9) and the complementation strain (CP4) to oleic acid (OA, 0.5%), Tween 20 (Tw20, 0.5%), calcofluor white (CFW, 200 µM) and Congo red (CR, 50 µM) on potato dextrose agar (PDA) or minimal medium (MM); (**B**) Quantitative analysis of chemical sensitivity on PDA. Significance of differences was analyzed using Tukey’s HSD post-hoc test. Means indicated by asterisks are significantly different from wild type, *p* < 0.01 (**), *p* <0.05 (*); (**C**) Percentage of growth reduction of *Δpex6* in relation to wild type grown on PDA amended with different concentrations of oleic acid for 3 days; (**D**) Nile red staining of fungal hyphae and conidia for the presence of lipid bodies showing blight red fluorescence.

**Figure 5 jof-06-00248-f005:**
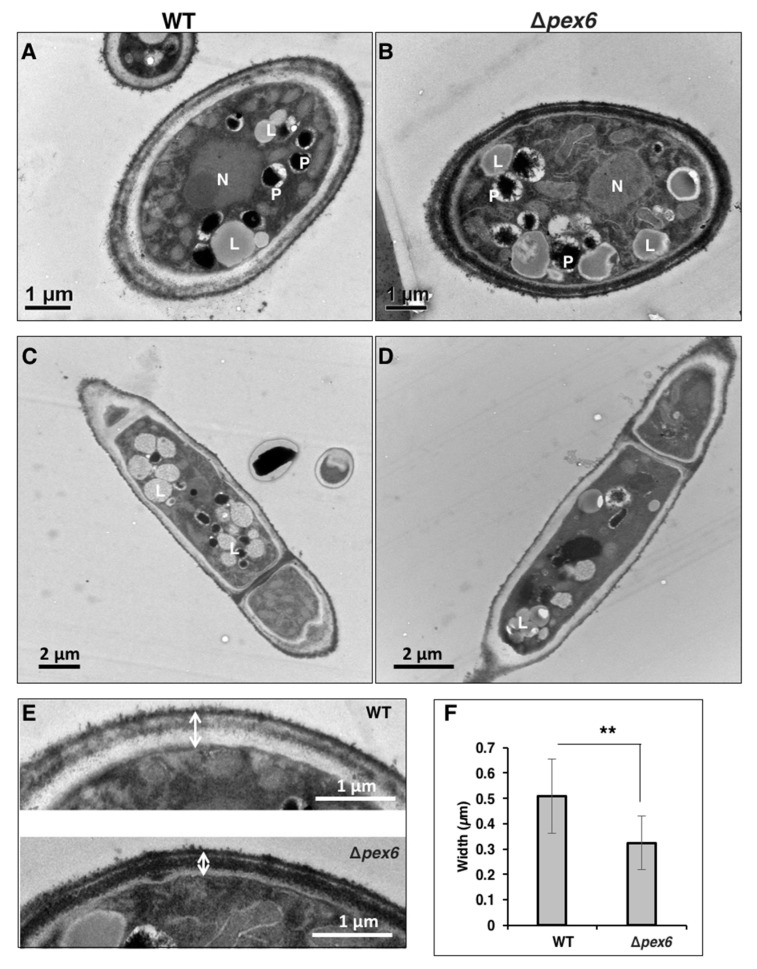
Transmission electron microscopy of *A. alternata* hyphae. (**A**) Cross section of a hypha of wild type (WT); (**B**) Cross section of a hypha of a *pex6* mutant (*Δpex6*); (**C**) Longitudinal section of a hypha of WT; (**D**) Longitudinal section of a hypha of *Δpex6*. Abbreviations: N, nucleus; P, peroxisome; L, lipid body; (**E**) Images of fungal cell walls; (**F**) Measurement of cell-wall thickness. Means (*n* = 30) indicated by asterisks are significantly different from one another, *p* < 0.01 (**).

**Figure 6 jof-06-00248-f006:**
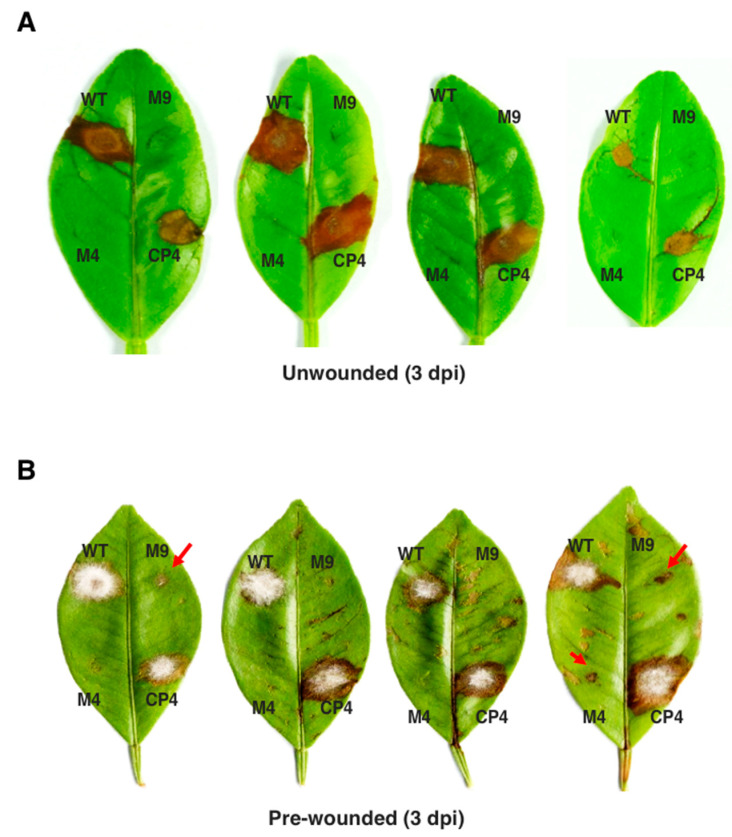
*Pex6* is required for *A. alternata* pathogenicity and lesion formation. (**A**) Fungal pathogenicity was assayed on detached calamondin leaves; (**B**) Fungal pathogenicity was assayed on citrus leaves that were wounded before inoculation. Each spot was inoculated with 5 µl of conidial suspensions (10^5^ conidia per milliliter) prepared from the wild-type (WT) two *pex6* mutants (*Δpex6*-M4 and M9) or the complementation strain (CP4). The leaves were maintained in a moisture chamber for lesion development. M4 and M9 induced much smaller lesions (indicated by red arrows). Each strain was tested on at least five different leaves and experiments were repeated at least three times.

**Figure 7 jof-06-00248-f007:**
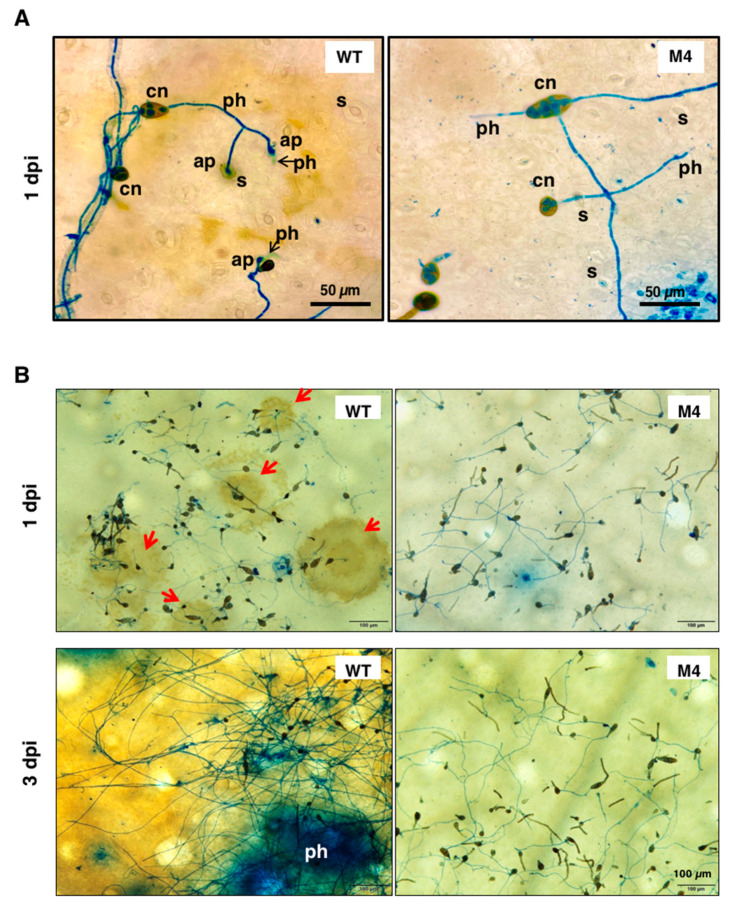
Light microscopy observation of conidia germination on the surface of calamondin leaves. (**A**) Conidia of wild type (WT) and the *Δpex6*-M4 mutant were placed on citrus leaves and observed by light microscopy, revealing the germination of conidia (cn), the formation of appressorium-like structures (ap) and the penetration of hyphae (ph) through stomata (s) after being staining with lactophenol cotton blue; (**B**) Microscopic lesions (indicated by red arrows) were observed on the surface of citrus leaves inoculated with wild-type conidia 1 day post inoculation (dpi) and coalesced to become large necrotic lesions 3 dpi. Microscopic lesions were rarely observed on citrus leaves inoculated with conidia collected from the *Δpex6*-M4 mutant.

**Figure 8 jof-06-00248-f008:**
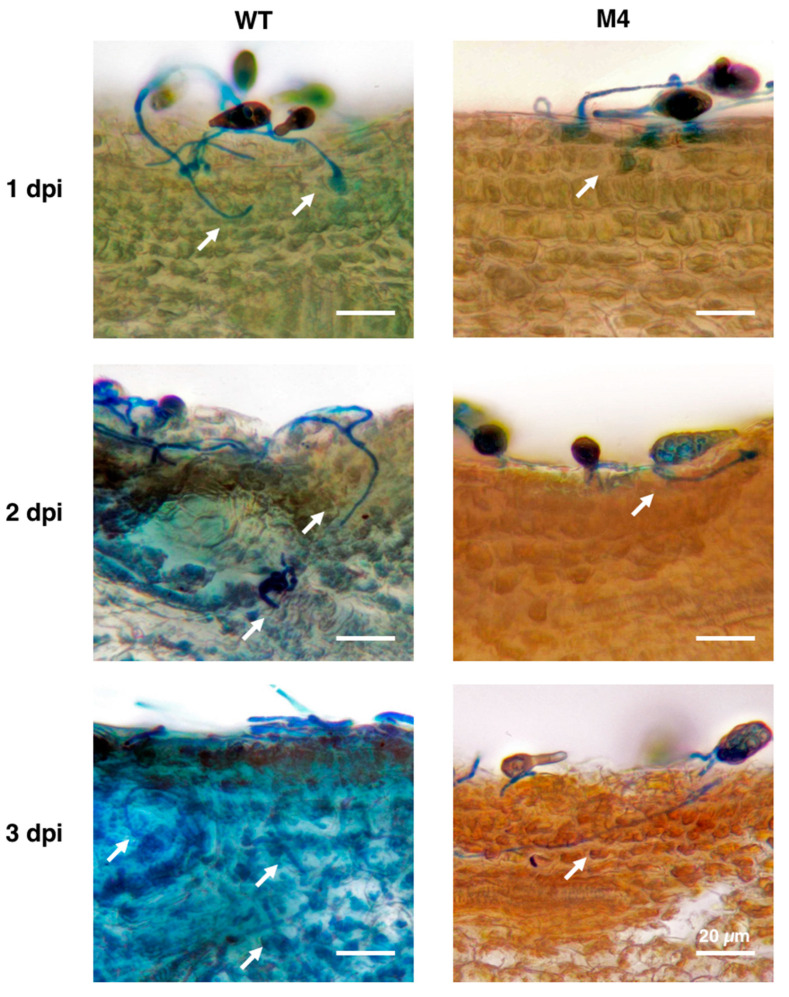
*Pex6* deficiency leads to penetration impairment. Calamondin leaves were inoculated with conidia prepared from wild type (WT) and the *Δpex6*-M4 mutant, thin sectioning by hands 1 to 3 days post inoculation (dpi), and observed by microscopy. Wild-type hyphae (indicated by white arrows) after lactophenol cotton blue staining were observed in the epidermal layer and mesophyll cells. Hyphae of the *Δpex6*-M4 mutant were observed only in the epidermal layer of citrus leaves. Bar = 20 µm.

**Figure 9 jof-06-00248-f009:**
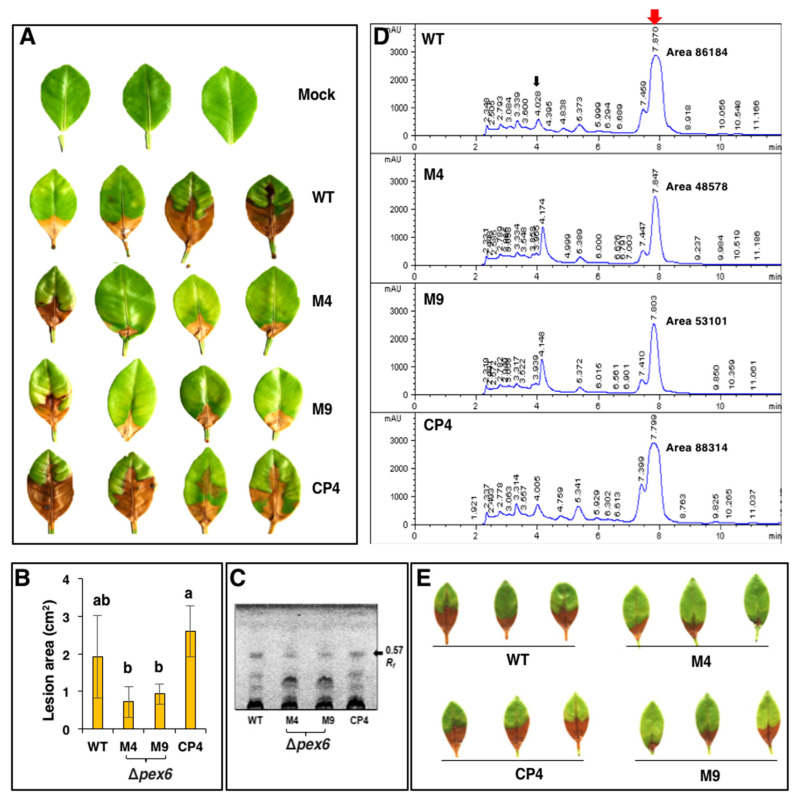
*Pex6* deficiency impacts toxin production in the tangerine pathotype of *A. alternata*. (**A**) A leaf necrosis assay for the toxicity of ACT toxin by soaking the petiole of the leaf in cell-free culture filtrates prepared from wild type (WT), two *pex6* mutants (*Δpex6*-M4 and M9) and the complementation strain (CP4) grown in potato dextrose broth for 7 days; (**B**) Quantitative measurement of necrotic lesions caused by cell-free culture filtrates (panel A) using Image J software. Significance of differences was analyzed using Tukey’s HSD post-hoc test (*p* ≤ 0.05). Means followed by the same letters are not significantly different; (**C**) Thin-layer chromatography (TLC) analysis of ACT; (**D**) High performance liquid chromatography (HPLC) analysis of ACT; (**E**) A leaf necrosis assay for the toxicity of ACT toxin eluted from HPLC (peaks with retention times of 7.4 and 7.9 min, indicated by a red arrow in panel **D**) by placing 5 µL elutes at the base of the petiole. The leaves were maintained in a moisture chamber for 3 days for lesion development. Each sample was tested on at least five different leaves and experiments were repeated at least three times.

**Figure 10 jof-06-00248-f010:**
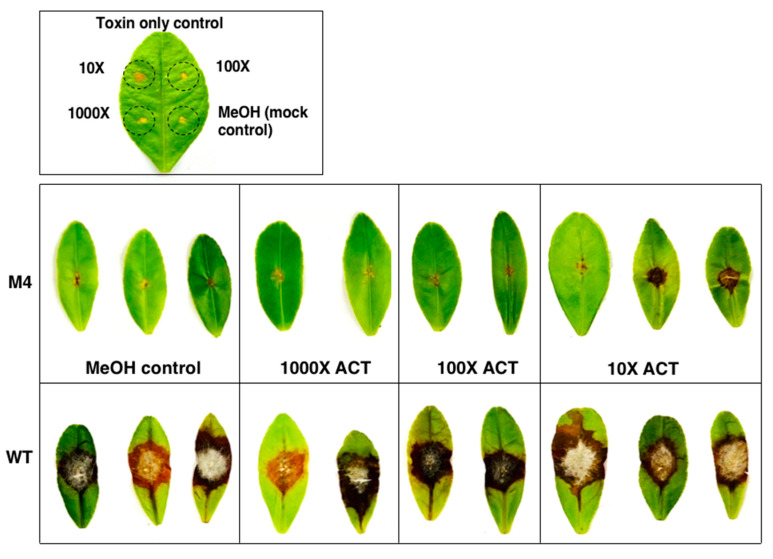
Partial restoration of *Δpex6* virulence with purified toxin. ACT (~12–15 µg toxin/g mycelium) was purified from culture filtrates of wild type grown in potato dextrose broth for 7 days using Amberlite XAD-2 resins and ethyl acetate, dissolved in methanol, diluted (10×, 100×, or 1000× in methanol), and applied onto calamondin leaves that were wounded before treatments. After 30 min, the leaves were inoculated with conidial suspensions prepared from wild type (WT) and the *Δpex6*-M4 mutant. The leaves were maintained in a moisture chamber for 3 days for lesion development.

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
