# Peer review of "Proper Functions of Peroxisomes Are Vital for Pathogenesis of Citrus Brown Spot Disease Caused by Alternaria alternata"

_jof, 2020, doi:10.3390/jof6040248_

Round 1

Reviewer 1 Report

In the manuscript “Proper functions of peroxisomes are vital for pathogenesis of citrus brown spot disease caused by Alternaria alternata”, the authors propose to make the functional characterization of the peroxin 6-coding gene (pex6). Overall the manuscript is well written and easy to follow. However, in my opinion the authors need to further clarify the following points prior to publication:

  • There is no definitive identification of the peaks. The ability to produce the necrotic spots is taken as the hallmark of the presence of this toxin, but that is also a feature sheared with other toxins? Is there other evidence that the authors can use to pinpoint these peaks as the ATC toxin? The authors should include other evidence such as the MS id (or at least approximate MW) or the qPCR of the toxin biosynthetic genes, if available.
  • Related to the previous point, authors need to clarify if in the toxin bioassays (lines 136-140) the levels of the toxin used were quantified? If different cell free extracts were used, please make clearer how the levels were normalized between batches? Similar in lines 152 - The peaks eluted in the different time intervals were polled together or used separately? How were the batches quantified and the incubation normalized?
  • Finally, the inoculation at the co-Inoculation of Δpex6 with toxin (lines) was done with cell free extracts or with the collected peaks? Which does a 10 fold dilution mean? The authors present the results for the controls without the conidial suspension, only in the top of the figure and say that this effect is slightly. In my opinion this is not enough for the authors conclude that the exogenous addition of the toxin restores, even if partially, the effect of the deletion.

MINOR:

Virulence assays – the conditions temperature, humidity and light/dark are missing in the materials and methods section:

Author Response

Response to Reviewer 1 Comments

In the manuscript “Proper functions of peroxisomes are vital for pathogenesis of citrus brown spot disease caused by Alternaria alternata”, the authors propose to make the functional characterization of the peroxin 6-coding gene (pex6). Overall the manuscript is well written and easy to follow. However, in my opinion the authors need to further clarify the following points prior to publication: [Response: We appreciate your helpful comments. We have made almost all of the revisions recommended by you. Please see our point-to-point responses below. I hope that these revisions are satisfactory. Thank you!]

  • There is no definitive identification of the peaks. The ability to produce the necrotic spots is taken as the hallmark of the presence of this toxin, but that is also a feature sheared with other toxins? Is there other evidence that the authors can use to pinpoint these peaks as the ATC toxin? The authors should include other evidence such as the MS id (or at least approximate MW) or the qPCR of the toxin biosynthetic genes, if available. [Response: The tangerine pathotype of alternata mainly produces ACT, which is toxic to susceptible citrus cultivars (Akimitsu et al., 2003). ACT was purified in accordance to the published protocol (Kohmoto et al., 1993). Its identity was also verified by LC-MS-MS in this study (data not shown). In addition, peaks eluted from HPLC also showed toxicity on calamondin leaves.]

  • Related to the previous point, authors need to clarify if in the toxin bioassays (lines 136-140) the levels of the toxin used were quantified? If different cell free extracts were used, please make clearer how the levels were normalized between batches? Similar in lines 152 - The peaks eluted in the different time intervals were polled together or used separately? How were the batches quantified and the incubation normalized? [Response: The amount of ACT was calculated and normalized to the dry weight of mycelium (see Lines 146-148 for detail). The 7.4/7.9 min peaks were co-eluted, dried, dissolved in 50-100 µl methanol (based on dry weight of mycelium), and tested for toxicity on citrus, showing necrotic lesions (Lines 347-349), while the 4.1 min peak failed to induce necrotic lesions.]

  • Finally, the inoculation at the co-Inoculation of Δpex6 with toxin (lines) was done with cell free extracts or with the collected peaks? Which does a 10 fold dilution mean? The authors present the results for the controls without the conidial suspension, only in the top of the figure and say that this effect is slightly. In my opinion this is not enough for the authors conclude that the exogenous addition of the toxin restores, even if partially, the effect of the deletion. [Response: ACT was purified from cell-free extracts with Amberlite XAD-2 resin and ethyl acetate as described by Kohmoto et al., (1993). ACT (~12-15 µg/g dry mycelium weight) was serially diluted (10X, 100X, 1000X) and applied to calamondin leaves 30 min before inoculation. ACT alone caused limited necrosis. Δpex6 alone also caused small lesions but the size of lesions increased in the presence of exogenous ACT. However, the lesions were much smaller than those induced by wild type. We, therefore, concluded that ACT could only partially restore Δpex6 virulence. The information is now clearly stated in the revision.]

MINOR:

Virulence assays – the conditions temperature, humidity and light/dark are missing in the materials and methods section: [Response: The information has been included in the revision (see Line 130 for detail).]

Reviewer 2 Report

In their paper, Wu et al. have deleted pex6, a gene conserved in Eukaryotes and involved in the peroxisome importomer. Previous studies in different fungi and especially in fungal plant pathogens have shown how important Pex6 and peroxisomes are for fungal physiology and especially pathogenesis. In their study, Wu et al. confirm this importance in the fungal pathogen Alternaria alternata tangerine pathotype. Wu et al. use different and powerful approaches to characterize the Pex6 mutant: standard and fluorescence microscopy, TEM, HPLC (toxin purification), pathogenesis tests, etc. Their results are robusts and convincing. However, on many aspects, this manuscript has to be improved. The presentation of some results has to be modified and improved and the introduction and the discussion on the role of Pex6 must be enriched with other previously published studies on Pex6 in fungal pathogens.

Minor points :

General concern:

Although the English is generally of a good level, it has to be improved and corrected.

All the photographs, and some of the diagrams must be nelarged for the future published version.

ΔPex6 must always be in italic as well as every gene or mutant cited in the text, it is sometimes lacking especially in the discussion.

L71: “hypothesized” typo

L17&19: English “ing” to “ed”?

L250: “redial” typo

L408: “per6” typo

Major concerns:

Introduction and discussion:

-The authors have deleted Pex6 in A. alternata tangerine pathotype and they have functionally characterized the mutant. Several of the phenotypes tested and several of the conclusions raised in the discussion have been previously reported in Ramos-pamplona et al, mol. Mic. 2006 in M. oryzae. I would like the authors ton include this important reference and consider their results with regards to this paper.

-Mat & Met

L100-102: the number of replicates for radial growth measurements is not indicated. Replicates are mandatory to allow the authors to show statistical significance to the comparisons made in the results, see below.

For the ΔPex6 deletion, it is not clear in the mat&met and in figure 1 if the entire pex6 CDS is replaced or if the Hph K7 is inserted inside the CDS. This has to be clarified in the text and in the figure.

Fig1: for the Southern blot on the left, there is signal detected above the 7.3 kB band in ΔPex6 strains. What does it correspond to ?

-Radial growth measurements:

Throughout the manuscript, the authors mention percentages of increase or of reduction in growth rate to show the different effects of the pex6 deletion in different conditions. A diagram including every measurements combined with statistical analyses (of replicates) is hugely needed in order to follow author’s interpretations. I recommend that this diagram indicates the actual growth rates rather than the ratio of growth rate compared to WT (I don’t see any inconveniency to have this figure in supplemental data). I suggest a second diagram indicating variations (% of reduction or increased) if the authors prefer this representation. The point here is that lots of arguments are provided by these growth rate analyses but they are hard to follow: different media are used (PDA and MM) and different components are tested on these media, some effects are minor other are major.

-Figure 2E: It is not clear if this diagram represents the percentages of appressorium-like on total conidia or on germinated conidia. What would be informative is the percentage of appressorium-like on germinated conidia. If this the diagram 2E represents percentages of appressorium-like on total conidia, this diagram is misleading and contradictory to what is indicated L226-227 where it is written that Δpex6 formed appressorium-like similarly as the WT. In addition, L305-306 are also in contradiction with L226-227.

Whether or not Δpex6 germinated conidia (germ tubes) produce less appressorium-like than the WT depending on the conditions (PDM medium, leaf, etc) must be clarified. A small table can help.

L228: I suggest the authors should speak of conidial viability rather than cell viability.

L234-235: what has been done to test the effect of Ca(NO3).4H2O is not clear and without a clear experiment with controls, the assumption on this toxic effect may not be included in the manuscript.

L244-255: This chapter requires the new diagram suggested above to be clearly interpreted. Especially the effect of oleic acid.

L254: the authors claim that ΔPex6 fails to efficiently utilize oleic acid but L245-248, they claim oleic acid (or tween) “partially restores Δpex6 growth deficiency” which are for me two contradictory assumptions. This is also in contradiction with the sentence L225. Different media are used (PDA of MM) and the relations between different media and oleic acid effects have to be clarified.

It has been shown in P. anserina that oleic acid can be toxic, I suggest the authors should consider this possibility for their interpretations.

-Figure 4C: This figure is supposed to show that there is less Nile red fluorescence signal in ΔPex6 mutants than in the WT and in CP4. However, in the red panel, every photographs are overexposed. Furthermore, they are more overexposed for the WT and CP4. Therefore, this figure cannot be interpreted as the evidence for less Nile red fluorescence in ΔPex6. This figure must be improved. I also suggest that the most apical part of the thallus should be chosen for comparison in addition to the fields of view shown here.

-Figure 5: In M. oryzae, melanin is absent from the appressorium cell wall in ΔPex6 mutants, can the author determine if it is the same for the A. alternata hyphal cell wall ?

-I suggest Figure 7 and 8 could be together

3.8 Pex6 Deficiency Impacts Toxin Production

My main concern in this chapter is the lack of normalization of ACT production/fungal biomass. Without normalization, since ΔPex6 growth rate is lower than WT, one can expect less fungal biomass and less toxin production in cultures. Hence, are the 40% less ACT content in ΔPex6 due to lower production in the mutant or is it due to lower fungal biomass producing the toxin in the culture before filtration?

On the same line, is the concentration of ACT (or any molecule eluted for each tested peak) in the different elutes controlled ? This has to be clarified. If not, the reduced effect of the toxin on leaves could simply be due to a lower amount in Δpex6 samples than in WT samples.

Finally, if the authors show a difference in activity of the ACT toxin depending on the genotype (and not on the quantity of toxin), the authors must discuss or explain what could explain this difference.

3.9: Co-inoculation of Pex6 with toxin Partially Restores Lesion Formation

Although the partial effect observed when 1/10 toxin content is added to the ΔPex6 strain for co-inoculation is convincing, the authors should comment on the concentration of the toxin in the assay compared to the concentration of the toxin during natural fungal infection. Are these comparable ?

-Discussion:

L383-L384: several mutants of pex import proteins such as Pex2 also affect peroxisome biogenesis in fungi. I would like the authors to be more specific to emphasize how pex6 in A. alternata may be more involved into import rather than biogenesis of peroxisomes. The authors must strengthen their argument. The same L427-L428.

L401: The mutant in F. graminearum was a RNAi mutant, it is therefore not surprising that its phenotypes may be somehow partial. The authors must consider it in the discussion.

L407: the authors claim that pex6 is dispensable for conidiation in A. alternata tangerine pathotype but they have not shown it in the paper. I would like the authors to present a quantification of conidiation in the A. alternata tangerine pathotype pex6 mutant to make this assumption.

L419-L421: As mentioned above, the effect of the calcium nitrate must be properly assayed for the authors to conclude on its toxic effects.

L440-441: This argument on oleic acid that promote growth in the WT must be supported by convincing and statistically analysed comparisons of growth rate.

L443: A. alternata unable to use glucose ?

Author Response

Response to Reviewer 2 Comments

In their paper, Wu et al. have deleted pex6, a gene conserved in Eukaryotes and involved in the peroxisome importomer. Previous studies in different fungi and especially in fungal plant pathogens have shown how important Pex6 and peroxisomes are for fungal physiology and especially pathogenesis. In their study, Wu et al. confirm this importance in the fungal pathogen Alternaria alternata tangerine pathotype. Wu et al. use different and powerful approaches to characterize the Pex6 mutant: standard and fluorescence microscopy, TEM, HPLC (toxin purification), pathogenesis tests, etc. Their results are robusts and convincing. However, on many aspects, this manuscript has to be improved. The presentation of some results has to be modified and improved and the introduction and the discussion on the role of Pex6 must be enriched with other previously published studies on Pex6 in fungal pathogens. [Response: We appreciate your helpful comments. We have made almost all of the revisions recommended by you. Please see our point-to-point responses below. I hope that these revisions are satisfactory. Thank you!]

Minor points:

General concern:

Although the English is generally of a good level, it has to be improved and corrected. [Response: Language has been improved for clarity, and all typos have been corrected.]

All the photographs, and some of the diagrams must be nelarged for the future published version. [Response: All photographs have been enlarged.]

ΔPex6 must always be in italic as well as every gene or mutant cited in the text, it is sometimes lacking especially in the discussion. [Response: Δpex6 has been italicized throughout the text.]

L71: “hypothesized” typo [Response: See Line 71.]

L17&19: English “ing” to “ed”? [Response: See Lines 17&19.]

L250: “redial” typo [Response: See Line 253.]

L408: “per6” typo [Response: See Line 415.]

Major concerns:

Introduction and discussion:

-The authors have deleted Pex6 in A. alternata tangerine pathotype and they have functionally characterized the mutant. Several of the phenotypes tested and several of the conclusions raised in the discussion have been previously reported in Ramos-pamplona et al, mol. Mic. 2006 in M. oryzae. I would like the authors ton include this important reference and consider their results with regards to this paper. [Response: The work done in M. oryzae by Ramos-Pamplona and Naqvi (Mol. Microbiol. 2006) has been included in the discussion (Line 421).]

-Mat & Met

L100-102: the number of replicates for radial growth measurements is not indicated. Replicates are mandatory to allow the authors to show statistical significance to the comparisons made in the results, see below. [Response: Each treatment contained three replicates and experiments were repeated at least three times (Lines 101-102).]

For the ΔPex6 deletion, it is not clear in the mat&met and in figure 1 if the entire pex6 CDS is replaced or if the Hph K7 is inserted inside the CDS. This has to be clarified in the text and in the figure. [Response: A 1.2-kb region of pex6 was replaced with a hyg cassette in the pex6 ORF (Lines 188-189, 199).]

Fig1: for the Southern blot on the left, there is signal detected above the 7.3 kB band in ΔPex6 strains. What does it correspond to ? [Response: A faint band (estimated >10 kb), likely due to incomplete digestion, was detected in all three samples (Line 201).]  

-Radial growth measurements:

Throughout the manuscript, the authors mention percentages of increase or of reduction in growth rate to show the different effects of the pex6 deletion in different conditions. A diagram including every measurements combined with statistical analyses (of replicates) is hugely needed in order to follow author’s interpretations. I recommend that this diagram indicates the actual growth rates rather than the ratio of growth rate compared to WT (I don’t see any inconveniency to have this figure in supplemental data). I suggest a second diagram indicating variations (% of reduction or increased) if the authors prefer this representation. The point here is that lots of arguments are provided by these growth rate analyses but they are hard to follow: different media are used (PDA and MM) and different components are tested on these media, some effects are minor other are major. [Response: Measurement of fungal growth is straightforward. Growth of Δpex6 was compared to that of wild type. Growth reduction in the treatment was compared to mock controls (PDA or MM alone). Only representatives were shown in Figure 4A. Per your suggestion, a new Figure 4B showing quantitative measurements of growth reduction is now included.]

-Figure 2E: It is not clear if this diagram represents the percentages of appressorium-like on total conidia or on germinated conidia. What would be informative is the percentage of appressorium-like on germinated conidia. If this the diagram 2E represents percentages of appressorium-like on total conidia, this diagram is misleading and contradictory to what is indicated L226-227 where it is written that Δpex6 formed appressorium-like similarly as the WT. In addition, L305-306 are also in contradiction with L226-227. [Response: Appressorium-like structures were determined among germ tubes not conidia (Lines 227-232). The structures were only observed on glass slides, 96-well microtiter plates with MM, and on the surface of leaves.]

Whether or not Δpex6 germinated conidia (germ tubes) produce less appressorium-like than the WT depending on the conditions (PDM medium, leaf, etc) must be clarified. A small table can help. [Response: Appressorium-like structures were determined among germ tubes not conidia. The structures were only observed on glass slides, 96-well microtiter plates with MM, and on the surface of leaves. Conidia of Δpex6 germinated poorly. When germinated on glass slides, microtiter plates or leaf surface, germ tubes of Δpex6 produced fewer appressorium-like structures in MM (Figure 2E).]

L228: I suggest the authors should speak of conidial viability rather than cell viability. [Response: The statement has been changed to “conidia viability” (Line 233).]

L234-235: what has been done to test the effect of Ca(NO3).4H2O is not clear and without a clear experiment with controls, the assumption on this toxic effect may not be included in the manuscript. [Response: We agree with your assessment. To avoid confusion, we have deleted the results associated with Ca(NO3).4H2O in the revision.]

L244-255: This chapter requires the new diagram suggested above to be clearly interpreted. Especially the effect of oleic acid. [Response: As shown in Figure 4A, oleic acid could enhance the growth of wild type and Δpex6. This was confirmed further by a quantitative analysis. A new Figure 4B showing quantitative measurements of growth reduction is now included.]

L254: the authors claim that ΔPex6 fails to efficiently utilize oleic acid but L245-248, they claim oleic acid (or tween) “partially restores Δpex6 growth deficiency” which are for me two contradictory assumptions. This is also in contradiction with the sentence L225. Different media are used (PDA of MM) and the relations between different media and oleic acid effects have to be clarified. [Response: Δpex6 grew poorly on MM containing glucose or oleic acid as the sole carbon source (Figure 4A). A combination of glucose and oleic acid could enhance Δpex6 growth. The growth enhancement by oleic acid is much drastic when Δpex6 was grown on PDA (Figure 4A). The effect of oleic acid on the growth of Δpex6 is dependent on medium (PDA vs. MM).]

It has been shown in P. anserina that oleic acid can be toxic, I suggest the authors should consider this possibility for their interpretations. [Response: No change. As shown in Figure 4A, oleic acid clearly enhanced the growth of both wild type and Δpex6. It is unlikely that oleic acid is toxic to A. alternata.]

-Figure 4C: This figure is supposed to show that there is less Nile red fluorescence signal in ΔPex6 mutants than in the WT and in CP4. However, in the red panel, every photographs are overexposed. Furthermore, they are more overexposed for the WT and CP4. Therefore, this figure cannot be interpreted as the evidence for less Nile red fluorescence in ΔPex6. This figure must be improved. I also suggest that the most apical part of the thallus should be chosen for comparison in addition to the fields of view shown here. [Response: Photos in Figure 4C are not over-exposed. WT contained ample of lipid bodies in both conidia and hyphae, showing strong red fluorescence after staining with Nile red. In contrast, lipid bodies were observed primarily in hyphae of Δpex6. Δpex6 hyphae contained fewer lipid bodies, showing much fainter red fluorescence than those of WT. This was confirmed further by TEM analysis (Figure 5).]

-Figure 5: In M. oryzae, melanin is absent from the appressorium cell wall in ΔPex6 mutants, can the author determine if it is the same for the A. alternata hyphal cell wall ? [Response: Melanin was purified from cell wall of wild type, Δpex6, and complementation strain, revealing no significant differences (data not shown).]

-I suggest Figure 7 and 8 could be together [Response: No change. You have previously suggested to enlarged all photographs.]

3.8 Pex6 Deficiency Impacts Toxin Production

  • My main concern in this chapter is the lack of normalization of ACT production/fungal biomass. Without normalization, since ΔPex6 growth rate is lower than WT, one can expect less fungal biomass and less toxin production in cultures. Hence, are the 40% less ACT content in ΔPex6 due to lower production in the mutant or is it due to lower fungal biomass producing the toxin in the culture before filtration? [Response: ACT was purified from cell-free extracts with Amberlite XAD-2 resin and ethyl acetate as described by Kohmoto et al., (1993). The amount of ACT was calculated and normalized to the dry weight of mycelium (see Lines 147-148 for detail).]
  • On the same line, is the concentration of ACT (or any molecule eluted for each tested peak) in the different elutes controlled ? This has to be clarified. If not, the reduced effect of the toxin on leaves could simply be due to a lower amount in Δpex6 samples than in WT samples. [Response: The amount of ACT was calculated and normalized to the dry weight of mycelium.]
  • Finally, if the authors show a difference in activity of the ACT toxin depending on the genotype (and not on the quantity of toxin), the authors must discuss or explain what could explain this difference. [Response: The amount of ACT was calculated and normalized to the dry weight of mycelium. ACT was dissolved in different volume of methanol based on fungal mass.]

3.9: Co-inoculation of Pex6 with toxin Partially Restores Lesion Formation

Although the partial effect observed when 1/10 toxin content is added to the ΔPex6 strain for co-inoculation is convincing, the authors should comment on the concentration of the toxin in the assay compared to the concentration of the toxin during natural fungal infection. Are these comparable ? [Response: ACT was purified from cell-free extracts with Amberlite XAD-2 resin and ethyl acetate. The amount of ACT was calculated and normalized to the dry weight of mycelium. The 7.4/7.9 min peaks were co-eluted, dried, dissolved in 50-100 µl methanol (based on dry weight of mycelium), and tested for toxicity on citrus, showing necrotic lesions, while the 4.1 min peak failed to induce necrotic lesions. ACT (~12-15 µg/g dry mycelium weight) was serially diluted (10X, 100X, 1000X) and applied to calamondin leaves 30 min before inoculation. ACT alone caused limited necrosis.  Δpex6 alone also caused small lesions but the size of lesions increased in the presence of exogenous ACT. However, the lesions were much smaller than those induced by wild type. We, therefore, concluded that ACT could only partially restore Δpex6 virulence. At similar fungal mass, WT and CP strains produce higher quantities of ACT than Δpex6 mutants. The information is now clearly stated in the revision.]

-Discussion:

L383-L384: several mutants of pex import proteins such as Pex2 also affect peroxisome biogenesis in fungi. I would like the authors to be more specific to emphasize how pex6 in A. alternata may be more involved into import rather than biogenesis of peroxisomes. The authors must strengthen their argument. The same L427-L428. [Response: TEM analysis revealed that peroxisomes were found in hyphae and conidia of both WT and Δpex6 with no obvious difference. Pex6, belonging to the members of the ATPase Associated (AAA) protein family, physically interacts with Pex1 and together assist Pex5 recycle back to the cytosol (Gould and Collins, 2002 Nature Rev. Mol. Cell Biol.) (Rucktäschel et al., 2011 Biochim. Biophys. Acta.). Based on TEM observations and literatures, we therefore have concluded that Pex6 plays nor roles in peroxisomal biogenesis.]

L401: The mutant in F. graminearum was a RNAi mutant, it is therefore not surprising that its phenotypes may be somehow partial. The authors must consider it in the discussion. [Response: Information regarding F. graminearum was omitted from the discussion section.]

L407: the authors claim that pex6 is dispensable for conidiation in A. alternata tangerine pathotype but they have not shown it in the paper. I would like the authors to present a quantification of conidiation in the A. alternata tangerine pathotype pex6 mutant to make this assumption. [Response: Because there is no difference in conidia formation by WT and Δpex6, the results are simply stated in the revision (Lines 210-212).]

L419-L421: As mentioned above, the effect of the calcium nitrate must be properly assayed for the authors to conclude on its toxic effects. [Response: We agree with your assessment. To avoid confusion, we have deleted the results and discussion associated with Ca(NO3).4H2O in the revision.]

L440-441: This argument on oleic acid that promote growth in the WT must be supported by convincing and statistically analysed comparisons of growth rate. [Response: As shown in Figure 4A, oleic acid could enhance the growth of wild type and Δpex6. This was confirmed further by a quantitative analysis. A new Figure 4B showing quantitative measurements of growth reduction with statistical analysis is now included.]

L443: A. alternata unable to use glucose? [Response: A. alternata utilizes glucose effectively while Δpex6 fails to use glucose as the sole carbon source.]

Round 2

Reviewer 2 Report

For this new version of the manuscript, the authors have replied to the issues raised one by one and convincingly. They have included some required changes. I therefore agree to accept the paper in the present form.